# Effects of Organic Fertilizer Addition to Vegetation and Soil Bacterial Communities in Saline–Alkali-Degraded Grassland with Photovoltaic Panels

**DOI:** 10.3390/plants13111491

**Published:** 2024-05-28

**Authors:** Aomei Jia, Zhenyin Bai, Liping Gong, Haixian Li, Zhenjian Bai, Mingjun Wang

**Affiliations:** 1Department of Grassland Science, College of Animal Science and Technology, Northeast Agricultural University, Harbin 150030, China; jiaaomm@163.com (A.J.); g069918029@163.com (L.G.); nmglhx@163.com (H.L.); baizj0306@126.com (Z.B.); 2The Research Center of Soil and Water Conservation and Ecological Environment, Chinese Academy of Sciences and Ministry of Education, Yangling, Xianyang 712100, China; zhenybai@163.com; 3Institute of Soil and Water Conservation, Chinese Academy of Sciences and Ministry of Water Resources, Yangling, Xianyang 712100, China; 4University of Chinese Academy of Sciences, Beijing 100049, China

**Keywords:** organic fertilizer, photovoltaic panels, Songnen grassland, saline–alkali degradation, soil bacteria

## Abstract

The Songnen grassland is an important resource for livestock production in China. Due to the intensification of anthropogenic activities in recent years, vegetation degradation has worsened, and the salinization of grassland has become increasingly serious, which severely affects the sustainable development of grassland animal husbandry. In this study, organic fertilizer addition was carried out at saline-and-alkaline-degraded Songnen grassland sites with photovoltaic panels, and we investigated the effects of organic fertilizer treatments on the vegetation and soil bacteria in these areas. The results showed that both organic fertilizer treatments increased the community composition and diversity indices of plants (*p* < 0.05); they also had significant effects on soil electrical conductivity and rapidly available potassium (*p* < 0.05). In the dominant phylum of bacteria, the relative abundance of *Firmicutes* increased without adding organic fertilizer under the photovoltaic panel; the addition of organic fertilizer had a significant effect on the relative abundance of Firmicutes and *Desulfobacterota* (*p* < 0.05), reducing their relative abundance, respectively. There were differences in the number of bacteria at the genus level under different treatments compared to the control, with the highest enrichment of bacteria occurring at the OFE position, and a significant difference (*p* < 0.05) being found between the control and the other four groups at the genus level of *g_norank_f_norank_o_Actinomarinales*. Organic fertilizer had a significant effect on the bacterial Simpson diversity index, with the most significant increasing trend found in OFE (the front eaves of the photovoltaic panel in fertilization area). The results of a correlation analysis showed that pH, electrical conductivity, and total nitrogen were the main factors affecting the soil bacterial community.

## 1. Introduction

Grasslands are the primary continental ecosystem, with a global area of 52.54 million km^2^, covering more than 40% of the global land area [1]. Grassland ecosystems are an important basis for animal husbandry and have various functions, such as supporting biodiversity, regulating climate, preventing soil erosion, and maintaining ecosystem balance [2,3]. In China, grasslands face serious challenges, including degradation, soil erosion, and declining productivity, due to natural factors and unsustainable land use patterns [4]. Soil salinization is the main cause of land degradation, which has serious consequences for global agricultural production and the ecological environment [5]. The western Songnen Plain is one of the three major areas of saline–alkali soil in the world; at present, the area of saline–alkali soil is more than 3 × 10^6^ hectares [6]. Due to environmental changes and human disturbances, the area of saline–alkali land is increasing, and the degree of salinization is increasing as well [7]. Salinization is an important factor driving the saline–alkali degradation of the Songnen grassland, seriously restricting its sustainable development. Therefore, it is necessary to implement grassland restoration measures to curb degradation and promote ecosystem restoration.

In recent years, considerable progress has been made in the development of renewable resources to optimize the existing energy infrastructure [8]. Among all renewable energy sources, the use of solar photovoltaic power is among the most promising technologies for sustainable energy generation [9,10]. At present, the global capacity of photovoltaic power generation accounts for about 1% of the total generated power. By 2030, it is expected to provide 15% of the global power supply, which is crucial for achieving global carbon neutrality [11,12]. Photovoltaic electric fields are usually located on reconstructed cultivated land and grasslands, which has a profound impact on the terrestrial ecosystem [13]. Previous studies have found that solar photovoltaic construction can affect the local climate and soil environment by disrupting the surface energy balance [14,15,16]. Photovoltaic panels can affect air humidity and soil water content by influencing the received photosynthetically active radiation (PAR) and by significantly reducing wind speed and turbulence [17,18,19,20]. These effects, in turn, affect plant communities, changing plant biomass, species richness, and vegetation cover [21,22].

Rational fertilization is important in the management of cultivated land and a necessary means to maintain the nutrient balance of grassland ecosystems [23]. Fertilization can be regulated to maintain a certain species composition, ensure an ideal yield, and balance the material input and output in the grassland ecosystem [24], thereby achieving sustainable grassland production [25,26]. An increasing number of studies focusing on the effects of organic fertilizers on grassland ecosystems have been conducted [27,28]. Organic fertilizers consist primarily of compounds naturally produced from artificially assisted crop residues or by-products through physical excretion or processing; their application improves soil health by gradually releasing nutrients into the soil and has been shown to be effective in improving the quality of saline soils [29,30]. The application of organic fertilizer can reduce soil acidification, increase soil urease and catalase activity, improve soil nutrient levels, and affect soil microbial community [31]. Organic fertilizer is rich in organic matter and beneficial microorganisms that can produce a large amount of organic acids and release slow-acting nitrogen, phosphorus, and potassium into the soil. Organic fertilizer can further effectively alleviate the rapid consumption of soil-available nutrients by plant growth and improve soil structure [32], thus affecting the diversity of soil microorganisms. PV panel construction and organic fertilizer application both play an important role in grassland restoration. However, it is still unclear how the combination of PV panel construction and fertilization affects grassland, so it is of great significance to study the addition of organic fertilizer to grassland areas with established PV panels.

In the past few decades, scholars have successfully implemented many restoration measures to alleviate degradation and increase vegetation cover in degraded grasslands [33,34]. At present, fencing is the most effective and economical measure to restore degraded grassland. Studies have shown that installing enclosures can effectively increase soil carbon, nitrogen, phosphorus storage, and plant biomass [35]. However, with the extensive and accelerated degradation of grassland, more effective ecological measures are required for the restoration of degraded grasslands. This study tested the effect of adding organic fertilizer to a degraded grassland area with photovoltaic panels in Songnen, which is important for maintaining the function of the grassland ecosystem. The objectives of this study are as follows: (1) explore the effects of adding organic fertilizers to degraded grassland areas in Songnen with photovoltaic panels on plant community composition and species diversity, soil properties, and bacterial community composition and diversity; and (2) clarify the intrinsic relationships between plant species diversity, soil properties, and changes in soil bacteria.

## 2. Materials and Methods

### 2.1. Study Site

The study site was in the photovoltaic electric field (N 45°46′11″, E 124°19′, 168 m above sea level) in Datong District, Daqing City, northern Songnen Plain. The study site has a typical northern temperate continental monsoon climate, with rain and heat in the same season, which is conducive to the growth of crops and pastures. The type of grassland in the study area is a temperate meadow steppe, with an average temperature of 4.2 °C, an extreme minimum temperature of −39.2 °C, and an extreme maximum temperature of 39.8 °C. The annual average frost-free period is 143 days. The annual precipitation is 427.5 mm. The main plant types are *Puccinellia tenuiflora* (Griseb.) Scribn., *Phragmites australis* (Cav.) Trin.ex Steud., *Suaeda glauca* Bge., *Leymus chinensis* (Trin.ex Bunge) *Tzvelev*, *Polygonum tortuosum* L., etc. The photovoltaic panels in the electric field are arranged in the determinant, with a height of 1.5 m in front of the board, a height of 4.5 m behind the board, a distance of 20 m between the boards, and all facing a north–south direction. At the time of the experiment, the photovoltaic panels had been present for two years. The photovoltaic area is enclosed, the grassland is not used for grazing, and no artificial interference is present. The construction of the photovoltaic power plant did not seriously interfere with the grassland vegetation and soil.

### 2.2. Experimental Design

On 8 July 2023, a sample plot was arranged in the vegetated part of the solar photovoltaic facility, and the whole photovoltaic power station was randomly divided into six areas. Of these six areas, three were selected to receive organic fertilizer in the form of pure sheep manure, produced by Bayannaoermeng Sheep Wetting Agriculture Co., Ltd., Inner Mongolia Autonomous Region, China. Moreover, three photovoltaic panels were randomly selected in each area, totaling 18 photovoltaic panels. In addition, a portion of grassland was selected around each photovoltaic panel as a control grassland; a total of nine control grasslands were selected. In view of the differences in temperature, light, soil temperature, and soil water content at different locations under the photovoltaic panel, quadrats were arranged in front of the eaves (FE) and right below the center (BP) of each photovoltaic panel, respectively. Fertilized locations in front of and under each photovoltaic panel were denoted as OFE and OBP, respectively; unfertilized locations in front of and under each photovoltaic panel were denoted as FE and BP, respectively; and the control grassland was referred to as CK (Figure 1). A total of 54 quadrats were set; the size of each quadrat was 1 × 1 m, and the fertilization rate was 1000 kg/ha. After one month, vegetation and soil samples were collected at various locations in the selected area, and field measurements were carried out.

### 2.3. Experimental Methods

#### 2.3.1. Vegetation Determination Method

Over the course of this study, all plant data were measured at the peak of forage growth at the end of July. The total coverage of each quadrat was determined, and the coverage, height, abundance, and frequency of all plant species in the quadrat were determined. Aboveground biomass (AGB) was measured by ground cutting and placed in marked paper bags. After drying at 65 °C for 48 h, the AGB of each species was weighed. The plant community was divided into four plant functional groups: perennial grasses (PG), perennial forbs (PF), annual and biennial herbs (ABH), and perennial semi-shrub-like herbs (PSH). In this study, the Shannon–Wiener diversity index, Simpson diversity index, and Pielou evenness index were used to characterize the α diversity of plant communities.

The Shannon–Wiener diversity index is calculated as follows:$$H^{\prime }=-\overset{S}{\underset{i=1}{\sum }}P_{i}{\mathrm{LnP}}_{i}$$

In the formula, S is the number of species in the quadrat, and Pi = Ni/N, where Ni is the number of individuals of species i and N is the total number of individual plants.

The Simpson diversity index is calculated as follows:$$D=1-\overset{S}{\underset{i=1}{\sum }}{\left(P_{i}\right)}^{2}$$
where Pi is the proportion of individual species, and i is the relative density of plant species (species density/total density of all species × 100).

The Pielou evenness index is calculated as follows:$$\mathrm{JP}=H^{\prime }\mathrm{Ln}\;S$$

#### 2.3.2. Soil Sampling and Analysis

After measuring the AGB parameters, the soil was sampled using a five-point method for each quadrat with a soil auger. Soil samples were collected at a depth of 0–20 cm. Five samples from the same square were fully mixed, quickly loaded into a sterile sealed bag, placed in an ice box for refrigeration, and quickly transported to the laboratory. The samples were divided into two parts, one part was air-dried, ground, and sieved to determine the physical and chemical properties of the soil. The rest was stored in a refrigerator at −80 °C for soil bacterial analysis.

Soil pH and electrical conductivity (EC; an important indicator of salinity) were determined in a 1:5 soil (air-dried)/water suspension [36]. Soil organic matter (SOM) was determined using the K_2_CrO_7_ volumetric method (external heating method) [37]. Total potassium (TK) and available potassium (AK) were determined via the flame photometric method [36]. Total phosphorus (TP) was determined using the NaOH fusion–molybdenum antimony anti-colorimetric method [36], and the available phosphorus (AP) was determined via the 0.5 M NaHCO_3_ (pH = 8.5) extraction–molybdenum antimony colorimetric method [36]. The total nitrogen (TN) and available nitrogen (AN) were determined by the Kjeldahl method and alkali solution diffusion method, respectively [38].

#### 2.3.3. Soil DNA Extraction, 16S rRNA Sequencing, and Data Analysis

According to the manufacturer’s requirements, total DNA was extracted from 0.25 g soil using the EZNA^®^ Soil DNA kit (Omega Bio-tek, Norcross, GA, USA). After DNA extraction, the quality of the extracted genomic DNA was measured using 1% agarose gel electrophoresis, and the DNA’s concentration and purity were determined using a NanoDrop2000 (Thermo Scientific, Waltham, MA, USA). Using the extracted DNA as a template, PCR amplification of the V3-V4 variable regions of the 16 S rRNA gene was performed using the upstream primer 338F(5′-ActCCTACGGgaggCAGCAGCAG-3′) and downstream primer 806R(5′-GGACTACHVGGGTWTCTAAT-3′) [39] carrying the barcode sequence. The PCR reaction system used is described as follows: 5 × TransStart FastPfu buffer 4 μL, 2.5 mM dNTPs 2 μL, upstream primer (5 μM) 0.8 μL, downstream primer (5 μM) 0.8 μL, TransStart FastPfu DNA polymerase 0.4 μL, Template DNA 10 ng, replenished to 20 μL. The amplification procedure was as follows: predenaturation at 95 °C for 3 min and 27 cycles (denaturation at 95 °C for 30 s, annealing at 55 °C for 30 s, extension at 72 °C for 30 s), stable extension at 72 °C for 10 min, and finally preservation at 4 °C (PCR instrument: ABI GeneAmp^®^ 9700, Thermo Fisher Technology Co., Ltd., Shanghai, China). PCR products were recovered using 2% agarose gel, purified using a DNA gel recovery and purification Kit (PCR Cleanup Kit, Over China), and quantified using Qubit 4.0 (Thermo Fisher Scientific, USA). The NEXTFLEX Rapid DNA-Seq Kit was used to construct a library of purified PCR products. Sequencing was performed using Illumina PE300/PE250 platforms (San Diego, CA, USA).

FLASH [40] (1.2.11) software was used for splicing. UPARSE v7.1 [41,42] software was used to perform operational taxonomic unit (OTU) clustering on the concatenated quality control sequences according to a 97% similarity. In order to minimize the impact of sequencing depth on the subsequent alpha diversity analysis, the sequence number of all samples was flattened to 20,000 (sequence flattening is recommended). After flattening, the average sequence coverage of each sample could still reach 99.09%. Taxonomic annotation of OTU species was performed using the RDP classifier [43] (version 2.11) for comparison with the Silva 16S rRNA gene database (v138) with a confidence threshold of 70%, and the community composition of each sample was calculated at different species classification levels. 16S function prediction analysis was performed using PICRUSt2 [44] (version 2.2.0) software.

### 2.4. Data Processing

Statistical analysis was performed using R (4.1.2) software. One-way analysis of variance (ANOVA) and Tukey’s HSD test were used to analyze the significant differences in plant diversity, soil properties, soil bacterial alpha diversity, and relative abundance of dominant phyla between groups. ANOVA was used to test the effects of the photovoltaic panels on plants, soil, and bacteria. Orijin 2021 software was used to map the plant community. Pearson’s correlation coefficient was calculated to analyze the correlation between vegetation, soil physical and chemical properties, and soil bacteria. The ‘vegdist’ software package in R (4.1.2) was used to perform RDA analysis on plant diversity, soil properties, and bacterial communities, and 999 alignment similarity analysis (ANOSIM) was performed to infer the potential relationships between bacterial community composition and plant diversity and soil properties.

## 3. Results

### 3.1. Effects of Organic Fertilizer Addition on Vegetation Community Composition and Diversity

The different restoration measures had no significant effect on the vegetation diversity index (*p* < 0.05). Compared with the control areas, the vegetation richness, Shannon–Wiener index, Simpson diversity index, and Pielou evenness index of the fertilized and unfertilized parts under the photovoltaic panels showed an overall increasing trend. The diversity index of OFE increased the most, and the diversity index of BP (the unfertilized area is directly below the photovoltaic panel) increased the least (Figure 2).

The PCA results showed that PCA1 and PCA2 explained 56.64% and 21.34% of the variation in plant community composition, respectively. There were significant differences in the plant community composition between different restoration measures and the control group (*p* < 0.05), indicating that different restoration measures changed the plant community composition and that the plant composition differed significantly across different locations under the photovoltaic panels (*p* < 0.05) (Figure 3 and Figure 4). There was no significant difference in total vegetation coverage and total aboveground biomass between the treatments under photovoltaic panels and organic fertilizer addition compared with CK (*p* < 0.05). However, the overall trend is upward, and the increase in OFE is greater; the PG coverage of FE (the front eaves of photovoltaic panels in unfertilized areas) was significantly different from that of CK, increasing by 125.3% (*p* < 0.05). The PF coverage of FE and BP was significantly different from that of CK, increasing by 85.5% and 87.2%, respectively *(p* < 0.05). The ABH coverage of OFE was increased significantly by 53.41% compared with CK (*p* < 0.05). There was no significant difference in PSH coverage, PG aboveground biomass, PF aboveground biomass, and PSH aboveground biomass compared with CK (*p* < 0.05). The ABH aboveground biomass of OFE was significantly increased by 53.71% (*p* < 0.05) compared with CK (Figure 4).

### 3.2. Effects of Organic Fertilizer Addition on Soil Physicochemical Properties

Except for the EC and AK, different remediation measures had no significant effect on soil physical and chemical properties (*p* < 0.05). Compared with the control plots, the pH, EC, AP, and AN content in different locations showed a downward trend, while the TK, TP, and TN contents showed an upward trend. 

For different locations under the PV panels, most soil properties were the highest under OFE (Table 1). The EC of OFE was significantly increased by 33.80% compared with CK (*p* < 0.05). The AK of BP was significantly increased by 5.26% compared with CK.

### 3.3. Effects of Organic Fertilizer Supplementation on Soil Bacterial Community Composition and Diversity

As shown by the calculated α diversity indices, different restoration measures have significant effects on the Simpson diversity of bacteria but have no significant effects on the Shannon uniformity, Chao1 richness, and Good’s coverage indices (*p* < 0.05) (Figure 5). Compared with the control plots, the bacterial α diversity index at the fertilization sites showed an overall increasing trend, and the increasing trend under the OFE treatment was the largest. The α diversity index of bacteria in the unfertilized part under the PV panel showed a decreasing trend, and the decreasing trend of BP was the largest.

At the phylum level, the relative abundance of the top 10 dominant bacteria at all sites was higher than 98% (*Actinobacteria*, 25.36%; *Proteobacteria*, 21.58%; *Acidobacteria*, 11.37%; *Chloroflexi*, 10.00%; *Gemmatimonadetes*, 7.46%; *Firmicutes*, 6.38%; *Myxobacteria*, 5.40%; *Bacteroidetes*, 4.48%; *Desulfobacterota* 3.90%; unclassified_k_norank_d_Bacteria, 2.61%). Compared with the controls, different restoration measures had a significant effect on the relative abundance of *Firmicutes* and *Desulfobacterota*, but they had no significant effect on other phyla (*p* < 0.05). The presence of photovoltaic panels increased the relative abundance of Firmicutes, and the relative abundance of Firmicutes in the FE plots was the highest. The addition of organic fertilizer under the photovoltaic panel partially reduced the relative abundance of *Firmicutes* and *Desulfobacterota*, among which the relative abundance of Firmicutes in OFE was the highest (Figure 6). The bacterial ASV measured at the genus level was 947, and the bacterial ASVs in CK, OFE, OBP (the fertilization area is directly below the photovoltaic panel), FE, and BP were 15, 33, 22, 11, and 14, respectively (Figure 7A). Compared with the control group, the number of bacterial ASV observed under the photovoltaic panels was significantly different. The examination of the differences between the groups showed that there was a significant difference in norank_f__norank_o__Actinomarinales between the control group and the other four groups at the genus level (Figure 7B).

### 3.4. Relationship between Organic Fertilizer Addition and Vegetation and Soil and Microbial Community

Through RDA analysis, the relationship between bacterial community composition, vegetation diversity, and soil properties was determined. The first two RDA1 and RDA2 axes explain 18.21% and 12.19% of the total variation in the bacterial community composition, respectively. The RDA results show that pH, EC, and TN had an effect on the bacterial community composition (*p* = 0.001; *p* = 0.00; *p* = 0.022) (Figure 8).

Pearson’s correlation analysis showed that *Actinobacillus* was positively correlated with TN, and Proteobacteria was negatively correlated with AK. Bacteroides were positively correlated with TK and TN and positively correlated with pH and EC. *Acidobacteria* was negatively correlated with pH and EC. *Chloroflexi* was negatively correlated with AN and negatively correlated with pH and EC. Unclassified_k_norank_d_Bacteria was negatively correlated with pH (Figure 9).

## 4. Discussion

### 4.1. Effects of Organic Fertilizer Addition on Vegetation and Soil Physicochemical Properties

The changes in plant community productivity and structure composition in the studied grassland were direct responses to the applied restoration measures. Studies have shown that organic fertilizer has a greater impact on grassland species richness, and the impact of organic fertilizer on species richness emphasizes that fertilization affects species richness by improving productivity [45,46]. The results of this study showed that both restoration measures applied increased plant richness, and the species richness of different locations in the fertilized plots increased more significantly, which was similar to the results of previous studies. Photovoltaic panel construction and organic fertilizer addition had no significant effect on plant diversity, and both restoration measures increased vegetation diversity, but the increasing trend of the fertilized part was larger. Among the observed increasing trends, that of the fertilized part in front of the eave of the photovoltaic panel was the largest, and the increasing trend of the unfertilized part of the photovoltaic panel was the smallest. Tong et al. [47] found that the application of organic fertilizer increased the Shannon–Wiener diversity index, Simpson dominance index, and Pielou evenness index. Wang et al. (2016) [48] found that the Shannon–Wiener and Simpson diversity indices under PV panels increased by 60% and 32%, respectively, compared with the control area. However, in typical grassland areas, there are fewer plant species and lower species diversity under photovoltaic panels [49,50]. This suggests that the effects of photovoltaic panels on plant diversity are varied in other regions and may even move in the opposite direction.

A large number of studies have shown that there is a correlation between biodiversity and biomass production [45]. The results of this study showed that the addition of organic fertilizer under photovoltaic panels increased the composition of the plant community. Photovoltaic panels reduce evapotranspiration, promote soil water retention, and promote the growth of PF and PG to a certain extent. Temperature variations also had different effects on the plant communities of the PF and PG plots. The results of this study show that there are differences in the total AGB of photovoltaic panels at different sites, where the total AGB at OFE increases most significantly and the total AGB at BP is lower than that at other sites. Greater precipitation is received at the front eave of photovoltaic panels, and the addition of organic fertilizer at this point can promote both water and resource availability and thus promote plant growth. Meanwhile, the low light level and litter layer at BP inhibit plant growth [46]. Grasses have shallow roots, which are usually distributed on the soil’s surface [51] and are sensitive to organic fertilizers, and applied fertilizers are thus quickly absorbed and utilized by these plants [52]. Weeds, in contrast, have deep roots and cannot easily absorb the applied organic fertilizers. In the study by Liu et al. (2019) [53], vegetation coverage under photovoltaic panels was enhanced, resulting in a decrease in solar radiation reaching the surface, effectively reducing soil temperature [54]. Temperature changes regulate interspecific relationships and promote the growth of different species, thus altering the overall composition of plant communities.

The growth of plants is inseparable from soil properties. Organic fertilizer promotes plant growth by increasing nutrient storage, the pH value, and the physical conditions of root growth by supplementing nutrients. Akhtar et al. (2018) [55] observed that the use of organic fertilizers significantly improved soil nutrient accumulation, including the accumulation of NPK, which was consistent with our study. This may be due to the fact that organic fertilizers are applied to the soil for a shorter period of time, resulting in an increase in humic acid concentration, soil aeration, and dilution of denser soil mineral fractions [56]. Choi et al. (2022) [14] found that reductions in soil C and N content may be caused by the removal of topsoil during the construction of photovoltaic arrays, and the soil texture may also be an important factor in how photovoltaic panels affect soil nutrient cycling, resulting in slightly higher soil nutrient contents in BP plots than in other sites under PV panels. Overall, due to the increase in vegetation coverage, biomass, and diversity, the soil quality indicators are gradually optimized, thus changing the composition of the plant communities.

### 4.2. Effect of Organic Fertilizer Addition on Soil Bacterial Community

The biodiversity and richness of microbial communities are considered to be critical to the integrity, functioning, and long-term sustainability of soil ecosystems, but they are usually altered by environmental disturbances [57]. Studies have shown that the application of organic fertilizer can significantly change the Shannon index and Simpson index of soil bacteria and increase functional diversity [58]. The results of this study showed that photovoltaic panel construction and organic fertilizer addition had no significant effects on the richness and diversity of soil bacterial communities, which may be due to the short fertilization time. Compared with CK, the bacterial diversity of OFE increased the most, while bacterial diversity in the BP position decreased, which may be due to the redistribution of precipitation caused by the direction of the photovoltaic panels.

Actinobacteria and Proteobacteria have been identified as the main bacterial communities in soils [59]. In this study, Actinobacteria and Proteobacteria were the most dominant phyla in all positions. Actinobacteria is a highly abundant phylum of bacteria, which uses lignin, hemicellulose, protein, cellulose, and other C/N substances to grow rapidly. Therefore, these bacteria need more C and N than other species [60]. This is similar to the results of this study, confirming that the appropriate addition of organic fertilizer is conducive to improving soil properties. Proteobacteria are fast-growing bacteria that can utilize a variety of carbon sources, which may be related to the extensive biodegradation and metabolic characteristics of Proteobacteria and their ability to inhabit various habitats [61]. It is observed that there are a large number of Firmicutes in the saline–alkali soil of the Songnen Plain in northeast China. Firmicutes and Proteobacteria are considered to be bacterial taxa with the ability to withstand and reduce saline–alkali stress [62]. The results of this study found that the addition of organic fertilizer had a significant effect on the relative abundance of Firmicutes, but had no significant effect on other dominant phyla, and there were significant differences in the dominant phyla at different locations under the photovoltaic panels. In OFE and FE plots, Firmicutes had the highest relative abundance, indicating that the position in front of the photovoltaic panel significantly changed the diversity and structure of the soil bacteria. The analysis of soil bacteria via a genus-level Venn diagram showed that, compared with the control group, there were differences in the number of bacteria in each location. Bacteria were most abundant in the OFE and OBP treatments. The abundance of g_norank_f_norank_o_Actinomarinales was significantly different at the generic level between the control group and the other four groups. These findings suggest that the addition of organic fertilizers changes the composition of microbial communities in the soil, possibly leading to the presence of beneficial bacteria that can reduce saline–alkali stress and increase soil nutrient levels [63].

### 4.3. Correlation Analysis between Organic Fertilizer Addition and Vegetation, Soil, and Soil Bacteria

Any change in environmental factors will change the soil bacterial community’s structure to a certain extent [64]. Soil bacteria can directly affect crop biomass by forming a symbiotic relationship with plant roots, changing their function on a single plant, or indirectly affecting crop biomass by changing the soil nutrient composition [65]. RDA correlation analysis showed that plant diversity was significantly correlated with bacterial diversity and composition. The addition of organic fertilizer to the photovoltaic panel area changed the composition of the plant community, thus changing the quality and quantity of root exudates and litter entering the soil, which, in turn, affected the composition and activity of the microbial community [66].

Plant diversity can directly affect soil bacterial diversity or indirectly affect soil bacterial diversity by affecting soil pH and salinity. pH and salinity (EC) are the main factors affecting the composition of the soil bacterial community [67,68], consistent with the RDA results of this study. Pearson correlation analysis showed that Bacteroides were positively correlated with pH and EC, while *Acidobacteria* and *Chloroflexi* were negatively correlated with pH and EC. With the increase in soil salinity, only a limited number of bacterial communities are able to withstand the enormous pressure of a high-salinity environment, resulting in changes in the relative abundance and community diversity of microorganisms [69]. Since most bacterial taxa have a relatively narrow growth tolerance, changes in soil pH may directly affect changes in bacterial communities. Bacteroides were significantly positively correlated with TK and TN, and *Chloroflexi* was significantly negatively correlated with AN. Akhtar et al. (2018) found that organic fertilizer significantly improved soil nutrient accumulation, including NPK, and our results confirmed this observation [55]. The effect of PV panel construction on grasslands is slightly different from that of organic fertilizer application, which may be due to changes to some important influencing factors on the ecosystem, including the leaching of sodium from PV panels, as well as precipitation, soil erosion, surface runoff, vegetation characteristics, and wind speed [70,71].

## 5. Conclusions

This experiment of adding organic fertilizer on saline–alkali-degraded grassland with photovoltaic panels showed that the two restoration measures increased plant diversity and community composition, improved the physical and chemical properties of soil, and directly and indirectly changed the diversity of soil bacteria. Among the dominant bacteria, the area without organic fertilizer under the photovoltaic panel increased the relative abundance of Firmicutes. The addition of organic fertilizer had a significant effect on the relative abundance of Firmicutes and *Desulfobacterota* (*p* < 0.05). The addition of organic fertilizer under the photovoltaic panel reduced the relative abundance of Firmicutes and *Desulfobacterota*. Compared with the control group, there were differences in the number of bacteria at the genus level under different treatments. Among them, the number of bacteria in the front eaves of the organic fertilizer area under the photovoltaic panel was the most enriched. There was a significant difference in g_norank_f_norank_o_Actinomarinales between the control group and the other four groups at the genus level. Correlation analysis showed that pH, EC, and TN were the main factors affecting the bacterial community composition. The results of this study help to further understand the impact of PV panel placement on the functioning of grassland ecosystems, highlighting the effect of organic fertilizer on the soil bacterial community, while also providing new insights into our understanding of bacterial biodiversity in organically fertilized soils. We emphasize that care should be taken when making grassland restoration decisions and that multiple factors should be considered in addition to achieving biodiversity and ecosystem service enhancement in order to achieve long-term grassland sustainability.

## Figures and Tables

**Figure 1 plants-13-01491-f001:**
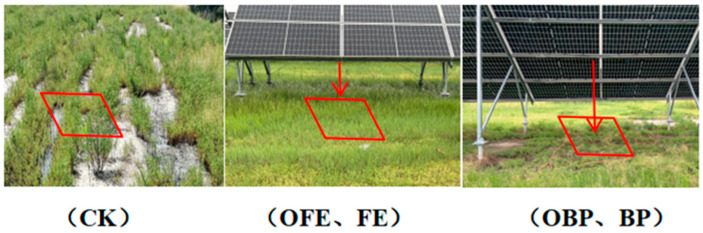
Different locations below the photovoltaic panel (CK: undisturbed grass around the photovoltaic panel; OFE: front edge of the fertilized part of the panel; FE: front edge of the unfertilized part of the panel; OBP: below the center of the fertilized section of the panel; BP: below the center of the unfertilized part of the panel).

**Figure 2 plants-13-01491-f002:**
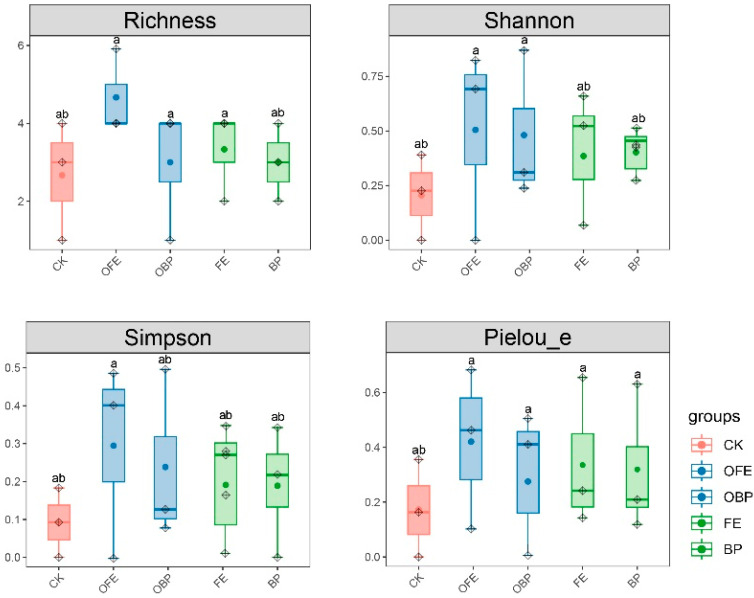
A box plot of vegetation alpha diversity index (CK: undisturbed grass around the photovoltaic panel; OFE: front edge of the fertilized part of the panel; FE: front edge of the unfertilized part of the panel; OBP: below the center of the fertilized section of the panel; BP: below the center of the unfertilized part of the panel. The midline of the box represents the median; the upper edge and the lower edge are the maximum and minimum values, respectively; the points outside the edge represent outliers). Note: Different letters mean significant differences between different positions (*p* < 0.05), the same letters mean no significant differences between different positions (*p* < 0.05).

**Figure 3 plants-13-01491-f003:**
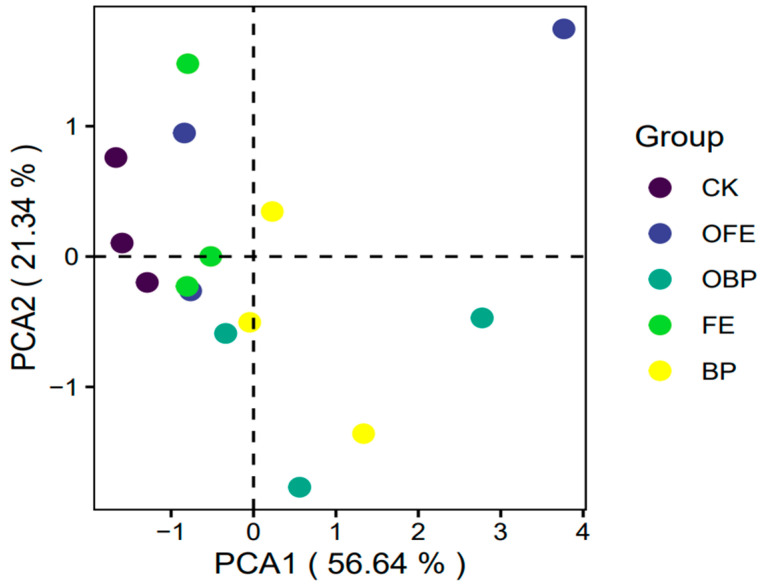
Principal coordinate analysis (PCA) of plant community composition at different positions under the photovoltaic panels (CK: undisturbed grass around the photovoltaic panel; OFE: front edge of the fertilized part of the panel; FE: front edge of the unfertilized part of the panel; OBP: below the center of the fertilized section of the panel; BP: below the center of the unfertilized part of the panel).

**Figure 4 plants-13-01491-f004:**
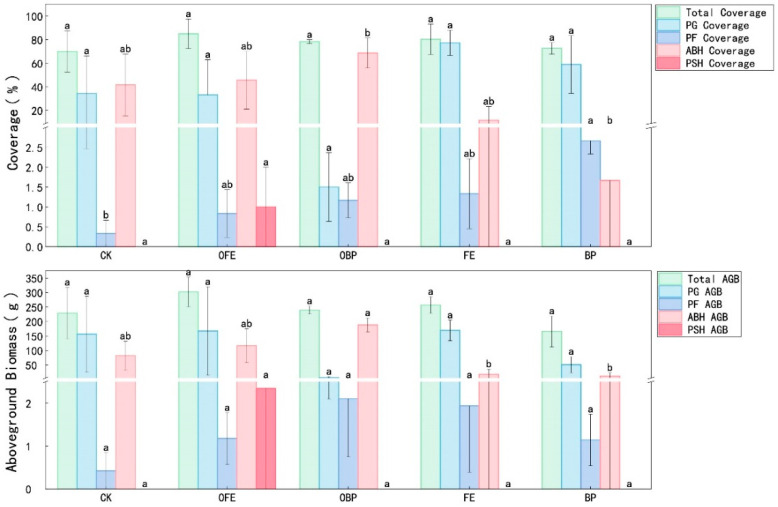
One-way ANOVA of the effects of different restoration measures on plant communities and functional groups (CK: undisturbed grass around the photovoltaic panel; OFE: front edge of the fertilized part of the panel; FE: front edge of the unfertilized part of the panel; OBP: below the center of the fertilized section of the panel; BP: below the center of the unfertilized part of the panel). Note: Different letters mean significant differences between different positions (*p* < 0.05), the same letters mean no significant differences between different positions (*p* < 0.05).

**Figure 5 plants-13-01491-f005:**
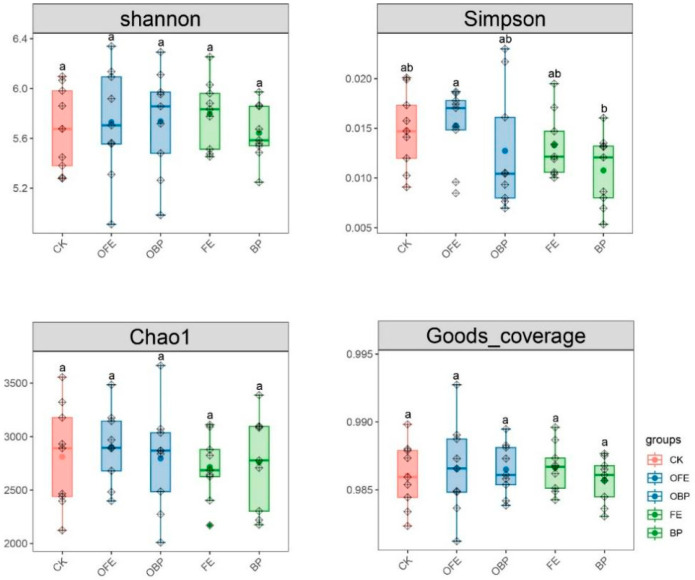
Box plot of soil bacteria alpha diversity index (CK: undisturbed grass around the photovoltaic panel; OFE: front edge of the fertilized part of the panel; FE: front edge of the unfertilized part of the panel; OBP: below the center of the fertilized section of the panel; BP: below the center of the unfertilized part of the panel. The midline of the box represents the median; the upper edge and the lower edge are the maximum and minimum values, respectively; and the points outside the edge represent outliers). Note: Different letters mean significant differences between different positions (*p* < 0.05), the same letters mean no significant differences between different positions (*p* < 0.05).

**Figure 6 plants-13-01491-f006:**
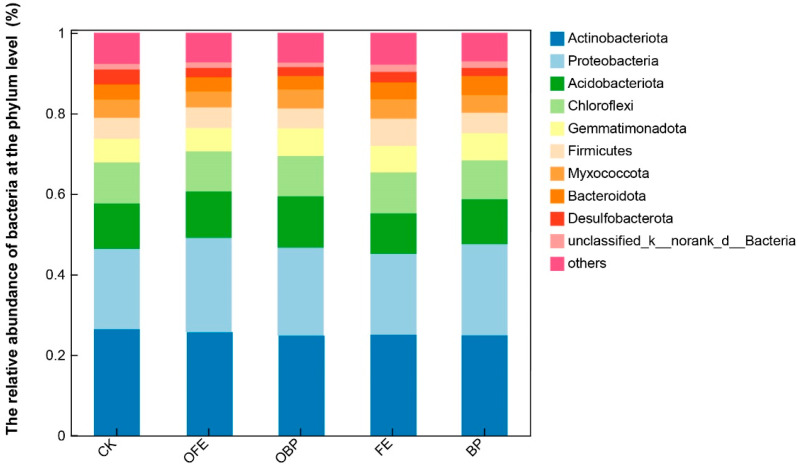
The relative abundance of bacteria at the phylum level (CK: undisturbed grass around the photovoltaic panel; OFE: front edge of the fertilized part of the panel; FE: front edge of the unfertilized part of the panel; OBP: below the center of the fertilized section of the panel; BP: below the center of the unfertilized part of the panel).

**Figure 7 plants-13-01491-f007:**
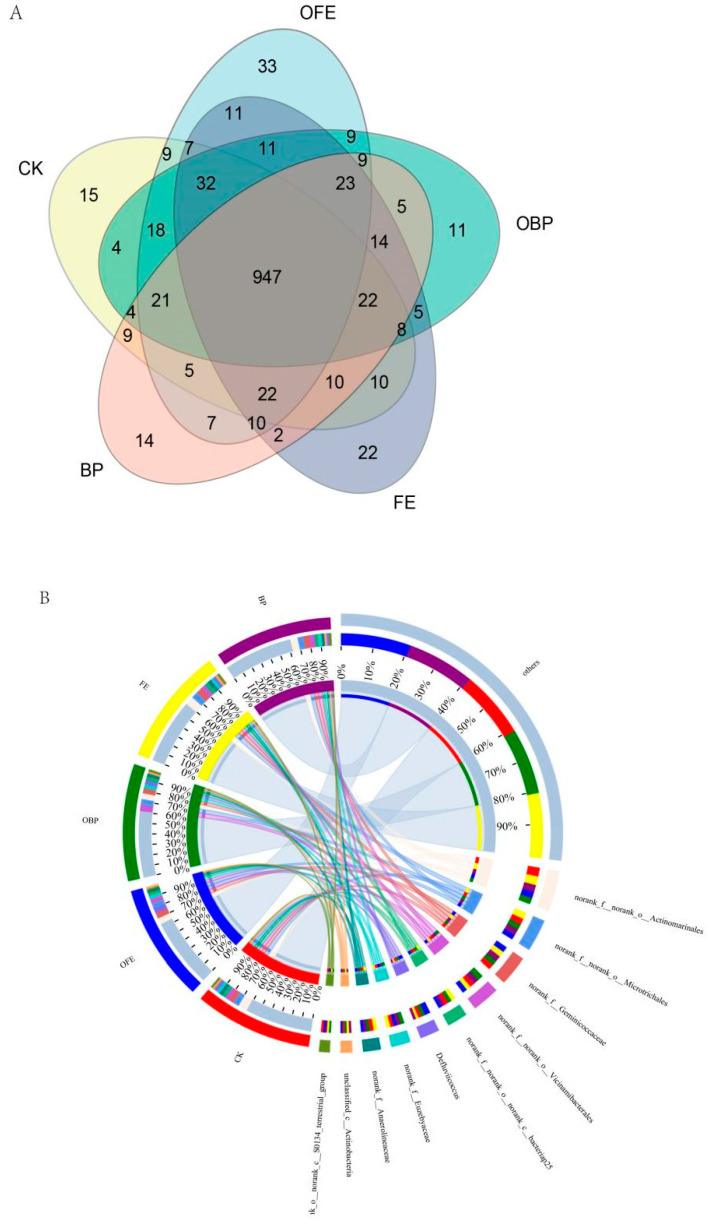
(**A**) Soil bacteria genus-level Venn diagram. (**B**) Soil bacteria genus-level Circos diagram (CK: undisturbed grass around the photovoltaic panel; OFE: front edge of the fertilized part of the panel; FE: front edge of the unfertilized part of the panel; OBP: below the center of the fertilized section of the panel; BP: below the center of the unfertilized part of the panel).

**Figure 8 plants-13-01491-f008:**
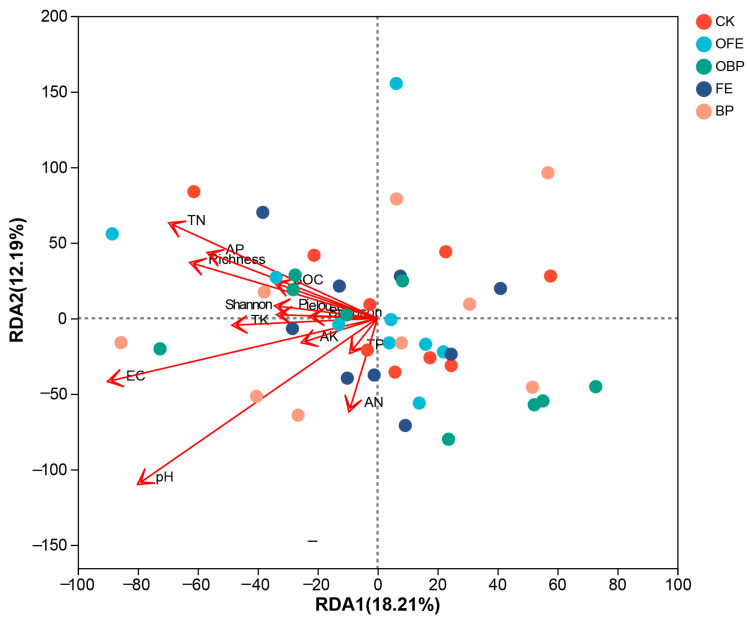
RDA analysis of soil bacteria and plant communities and soil physical and chemical properties under different restoration measures (CK: undisturbed grass around the photovoltaic panel; OFE: front edge of the fertilized part of the panel; FE: front edge of the unfertilized part of the panel; OBP: below the center of the fertilized section of the panel; BP: below the center of the unfertilized part of the panel).

**Figure 9 plants-13-01491-f009:**
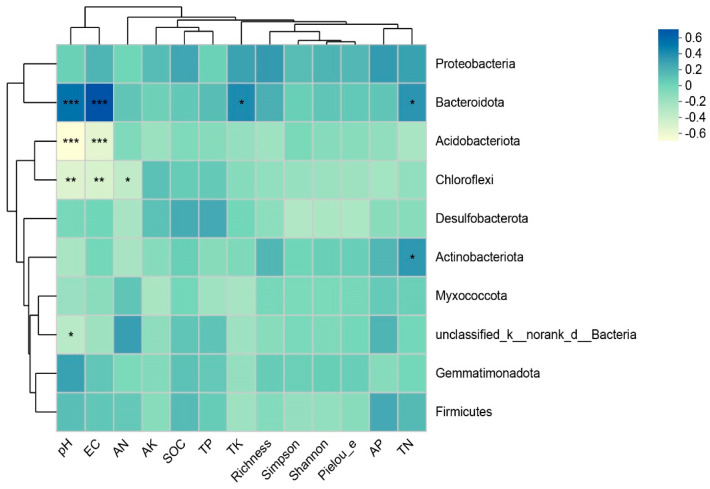
Pearson’s correlation analysis of soil bacteria dominant phyla with plant diversity and soil physical and chemical properties (CK: undisturbed grass around the photovoltaic panel; OFE: front edge of the fertilized part of the panel; FE: front edge of the unfertilized part of the panel; OBP: below the center of the fertilized section of the panel; BP: below the center of the unfertilized part of the panel). Note: * indicates significant correlation at the (*p* < 0.05) level, and **, *** indicates extremely significant correlation at the (*p* < 0.01) level, and no * indicates no significant association (*p* > 0. 05).

**Table 1 plants-13-01491-t001:** Analysis of the effects of different remediation measures on soil physical and chemical properties.

Name	CK	OFE	OBP	FE	BP	F	P
pH	9.46 ± 0.14 a	9.44 ± 0.13 a	9.41 ± 0.05 a	9.42 ± 0.08 a	9.12 ± 0.07 a	0.44	0.78
EC (mS/cm)	2.16 ± 0.33 ab	2.89 ± 0.43 a	1.89 ± 0.47 ab	1.91 ± 0.42 ab	1.25 ± 0.32 b	2.21	0.09
SOC (g/kg)	5.91 ± 2.67 a	6.80 ± 2.96 a	6.61 ± 3.51 a	6.30 ± 1.31 a	5.80 ± 0.85 a	0.23	0.92
TK (g/kg)	4.78 ± 0.97 a	5.31 ± 0.84 a	6.96 ± 3.11 a	6.87 ± 1.80 a	10.15 ± 3.52 a	1.71	0.17
AK (mg/kg)	142.85 ± 5.32 ab	140.79 ± 5.22 ab	132.44 ± 4.15 b	137.75 ± 6.54 ab	144.37 ± 4.62 a	0.81	0.52
TP (g/kg)	1.08 ± 0.04 a	1.27 ± 0.05 a	1.20 ± 0.10 a	1.14 ± 0.06 a	1.17 ± 0.08 a	0.88	0.48
AP (mg/kg)	149.57 ± 11.26 a	153.25 ± 9.82 a	137.18 ± 11.11 a	143.98 ± 9.37 a	148.50 ± 10.22 a	0.35	0.84
TN (g/kg)	0.99 ± 0.07 a	1.22 ± 0.14 a	1.12 ± 0.11 a	1.22 ± 0.10 a	1.02 ± 0.08 a	1.43	0.24
AN (mg/kg)	178.32 ± 12.59 a	176.45 ± 12.03 a	146.24 ± 7.43 a	151.68 ± 16.74 a	152.94 ± 15.54 a	1.30	0.29

Note: Different letters mean significant differences between different positions (*p* < 0.05), the same letters mean no significant differences between different positions (*p* < 0.05).CK: undisturbed grass around the photovoltaic panel; OFE: front edge of the fertilized part of the panel; FE: front edge of the unfertilized part of the panel; OBP: below the center of the fertilized section of the panel; BP: below the center of the unfertilized part of the panel.

## Data Availability

Data are contained within the article.

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
