# Peer review of "Effects of Organic Fertilizer Addition to Vegetation and Soil Bacterial Communities in Saline–Alkali-Degraded Grassland with Photovoltaic Panels"

_plants, 2024, doi:10.3390/plants13111491_

Round 1

Reviewer 1 Report

Comments and Suggestions for Authors

Jia et al. studied the effects of organic fertilizer on plant and soil bacterial communities in grassland with photovoltaic panels. They found that the organic fertilizer application increased the plant community composition, soil properties, and bacterial communities.

This manuscript needs significant improvement to be accepted for publication. I will only point out some major issues below and many minor issues need to be addressed too.

First of all, this article is not readable. It should move the material and methods section before the results.

Second, the experimental design lacks rationale. How many locations are included? It seems to me that it only had one location, and samples only have been collected from one-time point. It is difficult to draw any reasonable conclusions by such a design.

Third, the figures should be self-readable. However, all the legends are too concise to understand the figures. For instance, what are the ck, ofe, bp, fe, bp in the graphs?

The same issue for Table 1.

Furthermore, I doubt whether the authors did the PCR by themselves or not, it seems to me that the conditions in 4.3.3 did not make any sense. And I do not see people using 7% agarose gel at all.

Last, in the conclusion section, the thick-wall phylum was both increased and decreased in one sentence. I am not sure whether the authors read it before they submit it. 

Comments on the Quality of English Language

Jia et al. studied the effects of organic fertilizer on plant and soil bacterial communities in grassland with photovoltaic panels. They found that the organic fertilizer application increased the plant community composition, soil properties, and bacterial communities.

This manuscript needs significant improvement to be accepted for publication. I will only point out some major issues below and many minor issues need to be addressed too.

First of all, this article is not readable. It should move the material and methods section before the results.

Second, the experimental design lacks rationale. How many locations are included? It seems to me that it only had one location, and samples only have been collected from one-time point. It is difficult to draw any reasonable conclusions by such a design.

Third, the figures should be self-readable. However, all the legends are too concise to understand the figures. For instance, what are the ck, ofe, bp, fe, bp in the graphs?

The same issue for Table 1.

Furthermore, I doubt whether the authors did the PCR by themselves or not, it seems to me that the conditions in 4.3.3 did not make any sense. And I do not see people using 7% agarose gel at all.

Last, in the conclusion section, the thick-wall phylum was both increased and decreased in one sentence. I am not sure whether the authors read it before they submit it. 

Author Response

Dear reviewer,

Thank you for giving us "Effects of organic fertilizer addition on vegetation and soil bacterial communities in saline-alkali degraded grassland with photovoltaic panels "(ID: plants-2927429). These opinions are very valuable for the revision and improvement of our paper, and also have important guiding significance for our research. We have carefully studied your comments and made corrections, hoping for your approval.

1I feel very sorry for the negligence of the typesetting order. I have modified this part according to your opinion.

2.I am very sorry that you did not see the test site clearly because of my wrong writing. The experiment includes five sites, which are the undisturbed control position outside the photovoltaic panel area, the front eave of the panel and below the center of the panel where the photovoltaic panel is not fertilized, and the front eave of the panel and below the center of the panel where the photovoltaic panel is fertilized. The experiment really only sampled at one point in time, because the fertilization was applied in the same year and only treated for one year.

3.I have made corrections according to your comments. The positions and meanings of CK, OFE, OBP, FE and BP in the figure have been marked in the figure notes, and Table 1 has also been marked.The same issue for Table 1.

4.I have rewritten this section as you suggested and have changed to using 2% agarose gel.

5.I am very sorry for my wrong writing. I have revised the conclusion, and the corrected content is that the position of adding organic fertilizer under the photovoltaic panel reduces the relative abundance of actinomycetes, Aspergillus viridis and Firmicutes, and increases the relative abundance of Proteobacteria and Acidobacteria. The unfertilized location under the photovoltaic panel decreased the relative abundance of actinobacteria, Acidobacteria and Aspergillus viridis, while increased the relative abundance of Proteobacteria and firmicutes

Reviewer 2 Report

Comments and Suggestions for Authors

Overall, this is an interesting study conducted by Jia et al., to effects of organic fertilizer addition on vegetation and soil bacterial communities in saline-alkali degraded grassland with photovoltaic panels. I think the research question is interesting. However, the way the results are discussed in the manuscript needs minor rework. I hope the authors find these comments and suggestions helpful. I suggest minor revisions before this manuscript is acceptable for publication.  I have clearly indicated all my opinions and suggestions below.

Minor Comments:

There were no line numbers in the manuscript. I have indicated section numbers in my review.

What is the hypothesis of your research?

Section 1: You also need to justify/discuss the benefits of integrating vegetative diversity, degraded grass lands, soil fertility and photovoltaic panels. You have discussed these factors individually. Please consolidate them, especially in the last paragraph where you lay down your objectives.

Section 4.2:  Use metric units “1000kg/hm2”

Section 4.3.3.  “The threshold for all sequences was 0.005%.” What is this threshold?

Why did you use UNITE database? You are only looking at bacterial diversity, right?

“and the confidence interval for bacterial classification was 80%”. Please justify this sentence.

Figure 1: I see OFE having a significant higher richness value than other treatments. Could you please re-check the R-codes?

Section 2.3:

“Compared with the control, the α diversity index of some bacteria in organic fertilizer showed an overall upward trend, and the increase trend of OFE was the largest. The α diversity index of some bacteria in photovoltaic panels showed a downward trend as a whole, among which BP showed the greatest downward trend.”   

Where are these results? Is this from the α diversity?  α diversity is a measure of all bacterial diversity in your soil sample. Won’t be appropriate to use “diversity of some bacteria”

Figure 5: Please use a different color palette for this figure. It is difficult to differentiate the phyla abundance.

Section 3.1: Typos: “Weeds and weeds”

Comments on the Quality of English Language

Overall, this is an interesting study conducted by Jia et al., to effects of organic fertilizer addition on vegetation and soil bacterial communities in saline-alkali degraded grassland with photovoltaic panels. I think the research question is interesting. However, the way the results are discussed in the manuscript needs minor rework. I hope the authors find these comments and suggestions helpful. I suggest minor revisions before this manuscript is acceptable for publication.  I have clearly indicated all my opinions and suggestions below.

Minor Comments:

There were no line numbers in the manuscript. I have indicated section numbers in my review.

What is the hypothesis of your research?

Section 1: You also need to justify/discuss the benefits of integrating vegetative diversity, degraded grass lands, soil fertility and photovoltaic panels. You have discussed these factors individually. Please consolidate them, especially in the last paragraph where you lay down your objectives.

Section 4.2:  Use metric units “1000kg/hm2”

Section 4.3.3.  “The threshold for all sequences was 0.005%.” What is this threshold?

Why did you use UNITE database? You are only looking at bacterial diversity, right?

“and the confidence interval for bacterial classification was 80%”. Please justify this sentence.

Figure 1: I see OFE having a significant higher richness value than other treatments. Could you please re-check the R-codes?

Section 2.3:

“Compared with the control, the α diversity index of some bacteria in organic fertilizer showed an overall upward trend, and the increase trend of OFE was the largest. The α diversity index of some bacteria in photovoltaic panels showed a downward trend as a whole, among which BP showed the greatest downward trend.”   

Where are these results? Is this from the α diversity?  α diversity is a measure of all bacterial diversity in your soil sample. Won’t be appropriate to use “diversity of some bacteria”

Figure 5: Please use a different color palette for this figure. It is difficult to differentiate the phyla abundance.

Section 3.1: Typos: “Weeds and weeds”

Author Response

Dear reviewer,

Thank you for giving us "Effects of organic fertilizer addition on vegetation and soil bacterial communities in saline-alkali degraded grassland with photovoltaic panels "(ID: plants-2927429). These opinions are very valuable for the revision and improvement of our paper, and also have important guiding significance for our research. We have carefully studied your comments and made corrections, hoping for your approval.

  1. I have set the line number according to your suggestions, thank you for your guidance
  2. My research hypothesis is that laying photovoltaic panels and adding organic fertilizers have a positive impact on the restoration of degraded grassland in Songnen.
  3. I have modified this part according to your suggestion.
  4. I have modified the unit according to your opinion, and the revised one is 1000 kg/ha.
  5. I have modified and deleted this part according to your suggestions and combined with the report of the biological company
  6. I have deleted the UNITE database based on your comments and checked this section
  7. I have revised this section as you suggested, and according to the Bio company report, I have revised this section to: Annotate the OTU species taxonomy against the Silva 16S rRNA gene database (v138) with a confidence threshold of 70%, and count the community composition of each sample at different species classification levels.
  8. I have checked the R code according to your comments and modified this part
  9. I am very sorry for the trouble caused to you by my writing mistakes. I have checked and modified this part according to your comments. The "part of bacteria" in the article refers to the part under the photovoltaic panel that is fertilized and the part that is not fertilized, and I have modified it to different positions.
  10. I have modified the color matching of the abundance chart according to your suggestion. I am very sorry for the trouble caused to you
  11. I have corrected the typo "weeds and weeds" according to your suggestion.

Reviewer 3 Report

Comments and Suggestions for Authors

Reviewed manuscript presents quite new problem related to the photovoltaic panels instalation. These panels have frequently a  great area and are installed on the worse quality of soils, do not use to agricultural production. However, some treatments can  change chemical or/and physical soil features which led to improve its parameters.  In a consequence  is possible to change the conditions to soil  microorganisms  and plants  growth. Of course, panels themselves couse specific conditions to growth of plants in their proximity. So, the new studies related to relationships: panels- plants  or/and panels-soil parameters- microorganisms are very needed.

In relations to submitted manuscript I have only few questions:

1.       I understand that in this paper Authors presented only  1 year  results. However, experimental results are valid if they include at leat 2 years of experiment.

2.       In spite of presented purpose of these research my question is: if this experiment can have/have any utilitarian aims in aspect of panel locations on the worse soils covered with  grasses?

3.       If the composition of grasses species will be reacher in effect of organic fertilizer application and the soil parameters improve so , we can ask about the fate of the  grasses. It will be possible to find any utilization of these grasses growing during the vegetation period like grazing or as a component of the compost (with the wastes)?

4.       In spite of scientific aspect of this experiment it is worth to find  its practical utlization.

Author Response

Dear reviewer,

Thank you for giving us "Effects of organic fertilizer addition on vegetation and soil bacterial communities in saline-alkali degraded grassland with photovoltaic panels "(ID: plants-2927429). These opinions are very valuable for the revision and improvement of our paper, and also have important guiding significance for our research. We have carefully studied your comments and made corrections, hoping for your approval.

  1. Considering your suggestion, I would like to make the following explanation. The fertilization treatment in this experiment has only been carried out for one year, so we only took the experimental data of that year.
  2. Taking into account your suggestion, I have the following point of view to explain that the purpose of this experiment in the panel position on the poor soil covered with grass is to promote the restoration of degraded grassland and maintain the balance of grassland ecosystem.
  3. Taking into account your suggestions, I have the following views to explain that the use of these grasses can increase the plant diversity of degraded grasslands, and provide references for promoting the restoration of degraded grasslands and implementing grazing.
  4. Taking into account your suggestions, I have the following opinions to explain that the laying of photovoltaic panels and the addition of organic fertilizer under the panels are more effective ecological measures to promote the restoration of degraded grassland, and are of great significance to maintain the function of grassland ecosystem.

Round 2

Reviewer 1 Report

Comments and Suggestions for Authors

The manuscript has been significantly improved after revision. A few minor issues still need to be addressed, for instance:

In the Abstract, do not use the abbreviation of EC and AK. Please use the full name instead.

Line 73, change "are being conducted" to "have been conducted".

Line 104, "The research samples was collected" should be "The study site was"

Line 181, 16S rRNA sequencing

In all the figures and Tables, please add the description of different treatments. I noticed the authors changed in Figure 1, but not all others.

Comments on the Quality of English Language

It has been improved significantly. 

Author Response

Dear editor

First of all, thank you very much for taking time out of your busy schedule to read and revise my article. Thank you for your valuable suggestions. I have carefully studied your comments and made careful revisions to the paper according to the suggestions, as follows:

  1. I have changed the name of the indicator to the full name in the summary according to your suggestion
  2. I have changed "are being conducted" to "have been conducted" according to your suggestion.
  3. According to your suggestion, I have changed "The research samples was collected" to "The study site was".
  4. I have corrected the "16S rRNA sequencing" according to your suggestion.
  5. I have added the descriptions of different treatment methods in all the charts and tables according to your suggestions. Sorry for the trouble caused to your reading.
